# Effect of the Timber Legality Requirement System on Lumber Trade: Focusing on EUTR and Lacey Act

Ki-Dong Kim [1], Gyuhun Shim [1,2], Hyun-Im Choi [1] and Dong-Hyun Kim [1,*]

1    Division of Forest Management & Economics, National Institute of Forest Science,
     Seoul 02455, Republic of Korea; goldeast@korea.kr (K.-D.K.); qshim520@gmail.com (G.S.);
     hyunimchoi@gmail.com (H.-I.C.)
2    The London School of Economics and Political Science, London WC2A 2AE, UK
*    Correspondence: kimdh3165@korea.kr; Tel.: +82-2-961-2818

**Abstract:** This study provides novel insights into the policy effects of timber legality verification methods, specifically Due-diligence (under the European Union Timber Regulation (EUTR)) and Due-care (under the Lacey Act), on coniferous and non-coniferous lumber trade, highlighting their significance in the context of global lumber trade. Timber legality verification plays a pivotal role in the global timber trade. We comprehensively assess the impact of verification methods on coniferous and non-coniferous lumber trade, utilizing two decades of trade data (1997–2017) across approximately 160 countries. We employ the difference-in-differences method based on the gravity model of international trade, utilizing robust export–import data and demographic profiles. Our findings demonstrate that the effect of EUTR on coniferous lumber imports ranged between −0.32% and −0.05%, and that on non-coniferous lumber imports ranged between −0.44% and −0.05%, whereas the effect of the Lacey Act on coniferous lumber imports ranged between −0.93% and −0.09%. Non-coniferous lumber imports remained unaffected. The Voluntary Partnership Agreement (VPA) led to decreased exports to the EU and US. Our findings hold two key implications. First, Due-diligence exhibits more consistent policy effects than Due-care. Second, supporting VPA-participating countries is crucial for facilitating timber trade. These insights inform timber trade policies and sustainable practices.

**Keywords:** timber legality requirement system; lumber trade; VPA; gravity model; difference-in-differences

## 1. Introduction

The global production of timber faces a pressing dilemma with profound environmental, economic, and policy implications—the unrelenting destruction of forests and the alarming reduction in forested areas. This critical issue has far-reaching consequences, particularly in major timber-producing countries where forest degradation and deforestation are becoming increasingly dire concerns [1–3]. At the heart of this complex challenge lies the pervasive issue of illegal timber production, encompassing timber and products manufactured from timber that circumvent the legal regulations of their country of origin during production, distribution, and trade [4]. Moreover, excessive harvesting by entities holding logging permits further exacerbates the problem. The urgency of addressing illegal timber production cannot be overstated. It not only undermines the financial well-being of timber-producing nations by evading essential royalties, taxes, and financial obligations that logging companies must pay to the wood-producing country, but also exacts a heavy toll on ecosystems by contributing to forest degradation and endangering biodiversity [5]. One problem with illegal timber production is that it is interconnected with distribution and trade—logs produced illegally in timber-producing countries are exported or processed into timber products and exported to major importing countries, such as the US or Europe [6].

Recognizing the gravity of this issue, the international community, led by major timber-importing countries, has rallied around a shared imperative—the eradication of illegal timber imports. The 26th Conference of the Parties held in Glasgow, United Kingdom, in 2021 resulted in a summit declaration that underscored a collective commitment to combat deforestation, forest degradation, and forest restoration by 2030 [7]. Based on this international consensus, major timber-importing countries, such as the US and those in Europe, introduced a timber legality requirement system that prohibits the import of illegal timber to prevent forest loss and strengthen the functionality of this system.

The US amended the Lacey Act in 2008 to provide a legal basis for punishing companies or individuals engaged in the production or sale of illegal timber. Accordingly, the US prohibits the transportation, sale, retention, and export of illegally logged trees as well as other animal and plant resources. Additionally, the Lacey Act prohibits the distribution of illegal timber produced overseas throughout the US. Notably, it mandates that timber producers provide proof of the legal origins of their timber through the submission of requisite documentation to relevant authorities [8,9].

Meanwhile, the EU has banned the production and trade of illegal timber since 2003 and has established and implemented the Forest Law Enforcement, Governance, and Trade (FLEGT) action plan for sustainable forest management. Under the FLEGT, the EU introduced the European Union Timber Regulation (EUTR) in 2013, which restricts timber imports into the EU to those with verifiable legal provenance. Additionally, the EU actively engages in Voluntary Partnership Agreements (VPAs) with timber-producing countries to encourage legal timber production and export. These agreements entail comprehensive capacity-building efforts in participating nations to establish and operate effective timber legality verification systems [10]. In 2023, the EU implemented the European Union Deforestation Regulation (EUDR) to expand the scope of the regulation and its targeted countries from the EUTR as a part of the Glasgow Summit Declaration aimed at preventing deforestation and forest degradation.

Timber legality requirement systems can be divided into Due-care and Due-diligence. The fundamental distinction between these systems lies in the burden of proving timber legality, with Due-care placing this responsibility on producers and Due-diligence relying on a national-level verification system. Therefore, the choice of method to validate timber legality may yield varying policy outcomes.

The policy effects of the timber legality requirement system can be observed through changes in the volume of timber trade. Prior research in this area reported that the amendment of the Lacey Act did not affect the volume of timber and wood products in the US [11]. However, recent studies suggest that the introduction of the Lacey Act has negatively impacted the timber industry by reducing the import volumes of tropical wood and non-coniferous plywood and increasing their prices [12]. In addition, assessments of the EUTR's policy impact, employing the import-demand function of oak lumber, have generated inconclusive findings concerning its economic effects [13]. Additionally, research has highlighted the VPA's role in boosting Ghana's tropical log imports [14].

The timber legality requirement system entails the verification of legality throughout the entire supply chain of timber products. Therefore, not only countries producing timber, but also those involved in timber product processing, are exposed to the policy effects of the timber legality requirement system, reflected in the negative impact of this requirement on international log production. Notably, China, a pivotal hub for timber processing, has reported a decline in imports attributed to this policy effect [15].

The timber legality requirement system provides a powerful policy tool for curbing the import of illegal timber. To bolster and substantiate the implementation of this system dedicated to preventing the trade in illegal timber, a rigorous quantitative examination of its policy impact is imperative. Moreover, given the potential policy variations arising from distinct methods of proving timber legality, in this study, we aim to analyze the policy impact of the timber legality requirement system within the coniferous and non-coniferous lumber trade.

## 2. Characteristics of Each Type of Timber Legality System

The timber legality requirement systems of Due-care and Due-diligence, as adopted by the US and the EU, respectively, exhibit distinct characteristics in the verification of timber legality. Due-care places the onus on timber producers to substantiate the legality of their timber, underscoring their responsibilities and roles in the timber procurement process. Conversely, the EU's Due-diligence model delegates the verification of timber legitimacy to a third party, often an external expert or organization. This system prioritizes traceability and reliability throughout the timber importation process, including production, distribution, and trade, by scrutinizing all procedures from the initial stage of wood production through the entire supply chain [16,17].

The implementation of a timber legality requirement system requires time and financing [18]. In this context, Due-diligence can be a greater burden on timber-producing countries compared to Due-care. This is because Due-care determines the legality based on the relevant documents submitted by the producer. However, Due-diligence can provide comprehensive access to information related to the history of produced wood, a risk assessment of the overall wood supply chain, and additional information for risk mitigation or to monitor illegal activities. Consequently, Due-diligence necessitates the establishment of an organization capable of providing supplementary information on timber production or monitoring illicit activities [19].

Moreover, the EU has a VPA system, promoted as part of the 2003 EU FLEGT, which encourages and supports legal timber production in timber-producing countries [20]. Under this framework, the EU extends its backing to policy packages based on Due-diligence for VPA countries. The EU issues a FLEGT License for legally produced wood through the VPA, and timber bearing a FLEGT License is subsequently exported to other countries that require proof of timber legality, such as the EU and the US.

## 3. International Timber Trade Situation

Figure 1 shows the import trends of coniferous and non-coniferous lumber from 1990 to 2021, drawing data from the International Tropical Timber Organization (ITTO). The overarching trend reveals a consistent increase in international lumber imports. Notably, China's accession to the World Trade Organization in 2001 triggered a surge in lumber demand. However, the 2008 US financial crisis adversely impacted global demand, leading to a decline in lumber imports [21].

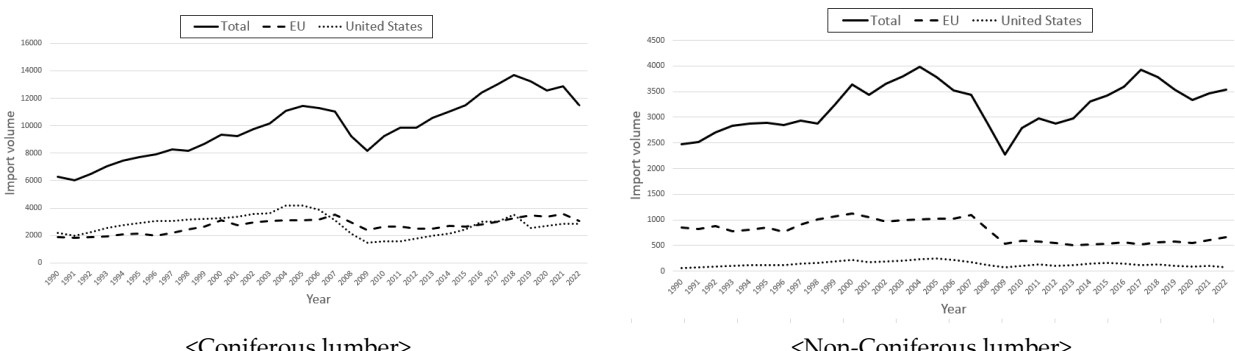

<Coniferous lumber>            <Non-Coniferous lumber>

**Figure 1.** Trends in lumber import volume during 1990–2021 (Unit: million m$^3$). Source: ITTO.

Subsequently, the US Federal Reserve Board implemented quantitative easing by lowering the base interest rate in response to the financial crisis. This policy decision prompted major countries to expand their liquidity, channeling ample money supply into the real estate market, thereby increasing construction demand.

On the international lumber supply front, Indonesia implemented a complete ban on log exports in 2001, followed by Russia's prohibition on log exports in 2003. These measures collectively contributed to an increase in lumber imports.



Meanwhile, the import volume declined in 2020 due to the coronavirus disease 2019 (COVID-19) pandemic, which caused shocks through maritime cargo volume, causing a temporary reduction in timber imports [22]. However, as major countries, including the US, engaged in expansive fiscal spending in response to the economic downturn caused by COVID-19, monetary liquidity increased in the market. In addition, the widespread use of telecommuting spurred the demand for housing. Consequently, a resurgence in lumber imports was observed, driven by both renovation efforts and the construction of new residences.

Figure 2 shows the trend of trade volume of timber and lumber by species in Ghana, Cameroon, Republic of Congo, and Indonesia, all of which are countries that ratified the VPA. On the left is the trend of coniferous log imports and lumber exports. The average annual import volume of coniferous timber was found to be 36,000 m$^3$. There were four periods when the import volume of logs increased rapidly. In particular, among the periods when imports of coniferous timber increased rapidly, 2014 marks the year that Indonesia passed the VPA. In addition, the average annual export volume of coniferous lumber is 66,000 m$^3$, and the highest export volume was in 2000, which was 252,000 m$^3$.

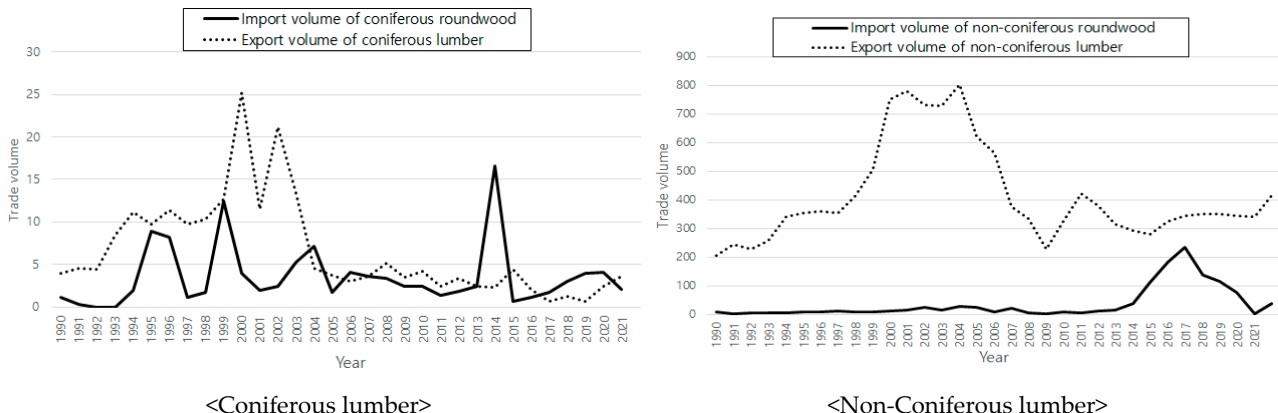

**Figure 2.** Trends in timber import and export volumes of VPA countries 1990–2021 (Unit: million m$^3$). Source: ITTO.

By comparison, the average annual import volume of non-coniferous timber is 371,000 m$^3$, and in 2017, the maximum of 2,334,000 m$^3$ was imported. The export volume of non-coniferous lumber averaged 4,141,000 m$^3$ per year, and has been on a continuous decline since 8033 m$^3$ was exported in 2004. However, in 2011, when Cameroon passed the VPA, it was found that 423 m$^3$ was exported.

Exports of non-coniferous lumber from VPA countries located in tropical regions were relatively higher than exports of coniferous lumber. However, it appears that coniferous lumber was also being exported. The supply chain regarding the export of coniferous lumber suggests that VPA countries import coniferous timber to export coniferous lumber. The reason for this is that the low wages of VPA countries who are developing countries located in tropical regions lead to low price competitiveness of coniferous lumber [23].

## 4. Methodology

The methodology employed in this study draws upon the gravity model, offering a comprehensive framework for analyzing the factors influencing international timber trade. Equation (1) encapsulates the essence of the gravity model, featuring the gravitational constant (G), gross domestic product (GDP) per capita (*GDP capita$_{i,j}$*), populations of the importing and exporting countries (*POP$_{i,j}$*), and trade distance (*Dist$_{i,j}$*) between those countries. GDP per capita and population are proxy variables for product purchasing power and the economic size of importing and exporting countries, respectively. The trade distance variable includes cultural heterogeneity as a component that hinders trade. As

trade distance increases, the time and cost of trade, as well as market uncertainty, increase because of changes in exchange rates and demand [24].

Although some prior studies leveraging the gravity model predominantly relied on gross domestic product(GDP) [25–27], in this study, we adopt a unique approach. We utilize population (POP) as a representation of economic size. However, given the high correlation between GDP and population within the model, which results in multicollinearity, GDP per capita is employed to overcome this issue. GDP per capita is calculated by dividing the GDP(GDP) by the population [28–30].

$$Trade_{(i,j)} = G \times \frac{GDP\ capita_{i,j} \times POP_{i,j}}{Dist_{i,j}}, \quad (i \neq j). \tag{1}$$

Equation (2) converts the gravity model into a log-linear form by taking the logarithm on both sides of Equation (1). This enables regression analysis using the ordinary least squares (OLS) method. The variable *dummy* is introduced to investigate factors influencing the trade of goods independently of the gravity variables [31,32]. In this study, the model is analyzed by combining the policy effect of the timber legality requirement system with the *dummy* variable's mechanisms among the variables of Equation (2) [33].

$$
\begin{aligned}
lnTrade_{i,j} = {} & \alpha_0 + \alpha_1 lnGDP\ capita_i + \alpha_2 lnGDP\ capita_j + \alpha_3 lnPOP_i \\
& + \alpha_4 lnPOP_j + \alpha_5 lnDist_{i,j} + \gamma dummy_{i,j}, \quad (i \neq j).
\end{aligned} \tag{2}
$$

In the difference-in-differences method, the treatment group is a group that participates in a particular policy, and the control group refers to the group to which the policy does not apply. This method compares the average change in the dependent variable over time between the treatment and control groups [34]. However, it is assumed that if the treatment group deviates from the application of the policy, it will show the same movement as the control group. Therefore, in this study, the treatment group ($treat_i$) is the lumber importing country that introduced the timber legality requirement system, and the time ($t_i$) is the year in which the timber legality requirement system was introduced. The policy effect ($\theta$) is an estimate of the interaction term, which is the interaction between the two aforementioned variables ($t_i \times treat_i$).

$$lnTrade_{i,j} = \alpha + \beta t_i + \gamma treat_i + \theta t_i \times treat_i + \epsilon_i. \tag{3}$$

Changes in the import volume of lumber are influenced not only by gravity variables and policy effects of the timber legality requirement system, but also by the state of the wood industry in the importing country. Therefore, we used the proportion of production in the construction industry ($GDPCons_i$) as a proxy variable representing the state of the construction industry, given that timber demand is derived from the construction industry. The policy effect of the timber legality requirement system was analyzed using Equation (4).

Equation (4) includes a gravity variable and a difference-in-differences term for the import volume of coniferous and non-coniferous lumber. In the difference-in-differences method, the implementation periods ($t_i$) of the system refer to the year 2008 when the Lacey Act was implemented and 2013 when the EUTR was implemented. The treatment group $\left(treat_{Policy,j}\right)$ represents the US, EU, and other countries that introduced the timber legality requirement system. The remaining importing countries excluding the US and EU were categorized into the control group. The policy effect of this system is expressed as the interaction term $\left(treat_{Policy,j} \times treat_{2008}\right)$, which is the interaction of the two afore-mentioned variables. As the timber legality requirement system in the US and the EU restricts the import of illegal timber, a negative sign is expected, signifying a reduction in lumber imports.

$$
\begin{aligned}
lnTrade_{i,j} = {} & \alpha_0 + \alpha_1 lnDist_{i,j} + \alpha_2 lnGDP\ capita_i + \alpha_3 lnGDP\ capita_j + \alpha_4 lnPOP_i + \alpha_5 lnPOP_j \\
& + \alpha_6 lnGDPCons_i + \beta treat_{Policy,j} + \gamma t_i + \rho treat_{Policy,j} \times t_i + \epsilon_{i,j}, \quad (i \neq j).
\end{aligned} \tag{4}
$$

As part of the FLEGT action, the EU operates a VPA to encourage trade in legally produced timber. Therefore, Equation (5) analyzes the simultaneous effect model, including variables reflecting the effect of the VPA based on the policy effect model of the timber legality system in Equation (4).

The VPA requires ratification by timber-producing countries, and even if ratification has been approved, a system is needed in which the Due-diligence of the VPA is reflected in the national system of the timber-producing countries. However, information about the time during which the VPA is fully implemented is unknown. Therefore, it was assumed that legal timber would be produced and exported based on the year in which the ratification of the VPA was passed.

Under this premise, Equation (5) treats the time when the ratification of the VPA countries took effect ($t_{VPAs}$), and the countries that signed the VPA, such as Ghana (GHA, 2010), Cameroon (CMR, 2011), Republic of Congo (COG, 2013), and Indonesia (IND, 2014), as the treatment group ($Treat_{i,VPAs}$), and the other countries as the control group. The policy effect of the VPA is determined by the interaction term between the treatment group and the period when the ratification took effect ($Treat_{i,VPAs} \times t_{VPAs}$). Therefore, the policy effect of the VPA is an estimate ($\theta$) that was evaluated for each aforementioned country.

The fundamental purpose of the VPA is to encourage the export of legally produced timber. Therefore, the coefficient is expected to be positively related to the import volume of coniferous and non-coniferous lumber, which are the dependent variables.

$$
\begin{aligned}
lnTrade_{i,j} = {}& \alpha_0 + \alpha_1 lnDist_{i,j} + \alpha_2 lnGDP\ capita_i + \alpha_3 lnGDP\ capita_j + \alpha_4 lnPOP_i + \alpha_5 lnPOP_j \\
& + \alpha_6 lnGDPCons_i + \beta_{policy} treat_{policy,j} + \gamma_{policy} t_i + \rho_{policy}\left(treat_{policy,j} \times t_i\right) \\
& + \sum \delta_{VPAs} Treat_{i,VPAs} + \sum \sigma_{VPAs} t_{VPAs} + \sum \theta_{VPAs}(Treat_{i,VPAs} \times t_{VPAs}) + \epsilon_{i,j},\ (i \neq j).
\end{aligned}
\tag{5}
$$

Three analytical methods are employed in this study: Pooled OLS with Robust Standard Error (RSE), Fixed Effect (FE) model, and Poisson Pseudo-Maximum Likelihood (PPML). In this study, RSE-Pooled OLS was applied because of RSE's ability to yield statistically significant estimated values when different variables are introduced into the model, thereby reducing the standard error's size [35]. However, in the timber trade, trading parties exhibit unobservable heterogeneity. The RSE-Pooled OLS falls short of adequately controlling this heterogeneity. Therefore, failure to control for the heterogeneity of each country causes heteroscedasticity. Consequently, this can lead to heteroskedasticity, where the model may fail to produce unbiased estimates if estimated with heteroskedasticity [36].

Heteroskedasticity caused by unobserved heterogeneity can be controlled using an FE model. A Fixed Effect (FE) model eliminates bias caused by unobservable confounders influencing the dependent variable under the assumption that such unobservable components are fixed over time [33]. However, time-invariant variables such as trade distance $\left(Dist_{i,j}\right)$ and the country implementing the timber legality requirement system ($treat_i$) are omitted from the estimation process in the country-pair FE analysis. In particular, in the gravity model, trade distance represents heterogeneity by country; however, this variable cannot be estimated using the FE model.

To address these limitations of the RSE-Pooled OLS and FE model, we incorporate the PPML analysis. The PPML method offers the advantage of controlling for heteroskedasticity and solving the problem of "0" trade, where observations have a "0" value [37,38].

## 5. Analysis Data

The analysis used data encompassing bilateral trade volumes of coniferous and non-coniferous lumber provided by the Food and Agricultural Organization of the United Nations (FAO; the data constructed by FAO are compiled from each country every August (Joint Forest Sector Questionnaire)). The FAO distinguishes between coniferous trees and non-coniferous (temperate and tropical) trees based on the HS code (coniferous timber HS codes: 440611 440691 440711 440712 440719; non-coniferous timber HS codes: 440612 440692 440721 440722 440725 440726 440727 440728 440729, 440791 440792 440793 440794 440795 440796 440797 440799) [39]. The data span from 1997 to 2017. The data are struc-

tured as panel data, which consists of 160 countries in bilateral trade relationships. In addition, variables related to the gravity model include trade distance, GDP per capita, and construction industry production. Data provided by CEPII and the World Bank were used.

Table 1 presents the descriptive statistics for coniferous and non-coniferous lumber-importing countries (referred to as "Report countries"). The EU, as the treatment group, exhibited an average import volume of 53,128 m$^3$. Conversely, the control group demonstrated an average imported volume of coniferous lumber at 23,015 m$^3$. A comparative analysis between the average import volumes of the control and treatment groups reveals that the EU's proportion of coniferous lumber imports is relatively larger. By contrast, the US, as part of the treatment group, reported an average import volume of coniferous lumber of 885,308 m$^3$, significantly higher than the control group's average of 29,732 m$^3$.

**Table 1.** Summary statistics of Report countries.

| | | Coniferous Lumber | | | | Non-Coniferous Lumber | | | |
| --- | --- | --- | --- | --- | --- | --- | --- | --- | --- |
| | | EU | | USA | | EU | | USA | |
| | | EUTR | Control | Lacey Act | Control | EUTR | Control | Lacey Act | Control |
| Import (*i*) (m$^3$) | Min. | 1 | 1 | 1 | 1 | 1 | 1 | 1 | 1 |
| | Max. | 4,421,148 | 11,523,000 | 47,781,561 | 11,523,000 | 7,322,700 | 14,927,115 | 1,207,059 | 14,702,351 |
| | Mean | 53,128 | 23,015 | 885,308 | 29,732 | 7098 | 7010 | 16,266 | 8303 |
| | St. dev. | 208,405 | 204,620 | 5,181,151 | 272,793 | 60,006 | 128,216 | 92,192 | 117,053 |
| Dist (*i,j*) (km) | Min. | 60 | 60 | 548 | 86 | 60 | 60 | 548 | 86 |
| | Max. | 19,586 | 19,772 | 16,180 | 19,772 | 19,586 | 19,772 | 16,180 | 19,772 |
| | Mean | 3545 | 5218 | 8037 | 6146 | 4790 | 5941 | 8838 | 6744 |
| | St. dev. | 3963 | 4466 | 4137 | 4823 | 4012 | 4569 | 4089 | 4670 |
| GDP per capita (*i*) (USD/capita) | Min. | 11,526 | 112 | 31,459 | 119 | 11,526 | 101 | 31,459 | 101 |
| | Max. | 123,679 | 102,913 | 60,110 | 102,913 | 123,679 | 102,913 | 60,110 | 102,913 |
| | Mean | 37,180 | 14,707 | 44,193 | 13,655 | 37,029 | 14,863 | 45,239 | 14,541 |
| | St. dev. | 16,108 | 18,134 | 8338 | 17,131 | 15,192 | 18,099 | 8646 | 17,713 |
| POP (*i*) (million) | Min. | 44 | 7 | 27,266 | 7 | 44 | 7 | 27,266 | 7 |
| | Max. | 8266 | 139,622 | 32,512 | 139,622 | 8266 | 139,622 | 32,512 | 139,622 |
| | Mean | 3252 | 9730 | 29,778 | 11,195 | 3253 | 11,273 | 29,981 | 12,509 |
| | St. dev. | 2772 | 28,216 | 1587 | 30,713 | 2783 | 30,462 | 1636 | 32,492 |
| GDP Cons (*i*) (%) | Min. | 10.4 | 3.2 | 18.0 | 4.6 | 10.4 | 4.1 | 18.0 | 4.1 |
| | Max. | 38.2 | 86.7 | 23.1 | 86.7 | 38.2 | 86.7 | 23.1 | 86.7 |
| | Mean | 22.8 | 31.1 | 20.7 | 31.4 | 22.9 | 30.7 | 20.5 | 30.7 |
| | St. dev. | 4.3 | 11.7 | 1.5 | 11.9 | 4.1 | 10.7 | 1.5 | 10.8 |

In terms of non-coniferous lumber, the EU's average import volume was 7098 m$^3$, which was not significantly different from the control group's average of 7010 m$^3$. However, the US, as part of the treatment group, reported an average import volume of 16,266 m$^3$, nearly twice as large as that of the control group, which stood at 8303 m$^3$.

Table 2 delves into the descriptive statistics of exporting countries (referred to as "Partner countries"). In terms of coniferous lumber, the average export volume of VPA countries to the EU was 197 m$^3$, lower than the average of 2073 m$^3$ for the control group. This trend can also be observed in the volume of exports to the US.

**Table 2.** Summary statistics of Partner countries.

| | | Coniferous Lumber | | | | Non-Coniferous Lumber | | | |
| | | EU | | USA | | EU | | USA | |
| | | EUTR | Control | Lacey Act | Control | EUTR | Control | Lacey Act | Control |
|---|---|---|---|---|---|---|---|---|---|
| Export (*j*) (m³) | Min. | 1 | 1 | 17 | 1 | 2 | 2 | 20 | 1 |
| | Max. | 9217 | 113,312 | 394 | 113,312 | 329,000 | 1,456,142 | 319,131 | 1,456,142 |
| | Mean | 197 | 2073 | 98 | 2153 | 12,065 | 8119 | 20,335 | 8028 |
| | St. dev. | 619 | 10,684 | 101 | 10,786 | 27,604 | 63,931 | 46,328 | 63,575 |
| Dist (*i,j*) (km) | Min. | 3800 | 190 | 8246 | 190 | 3800 | 190 | 8246 | 1 |
| | Max. | 12,188 | 19,772 | 16,180 | 19,772 | 12,679 | 19,772 | 16,180 | 19,772 |
| | Mean | 7493 | 6872 | 11,587 | 7031 | 6883 | 7240 | 11,170 | 7235 |
| | St. dev. | 3139 | 3746 | 3719 | 3837 | 2652 | 3674 | 3165 | 3677 |
| GDP capita (*j*) (USD/capita) | Min. | 258 | 258 | 1217 | 258 | 258 | 258 | 258 | 258 |
| | Max. | 3884 | 3923 | 3643 | 3923 | 3923 | 3923 | 3884 | 3923 |
| | Mean | 1268 | 1665 | 2202 | 1632 | 1596 | 1730 | 1579 | 1730 |
| | St. dev. | 852 | 983 | 883 | 984 | 1006 | 1010 | 1031 | 1010 |
| POP (*j*) (million) | Min. | 287 | 313 | 2034 | 313 | 287 | 287 | 287 | 287 |
| | Max. | 26,465 | 26,465 | 26,156 | 26,465 | 26,465 | 26,465 | 26,465 | 26,465 |
| | Mean | 10,162 | 11,731 | 10,979 | 11,642 | 7219 | 7746 | 7479 | 7746 |
| | St. dev. | 10,164 | 10,553 | 11,165 | 10,530 | 9583 | 9755 | 9830 | 9755 |

By contrast, the average export volume of coniferous lumber of VPA countries to the EU was 12,065 m³, while the exports of the control group to the EU were 8199 m³, indicating a relatively high proportion of exports to the EU. However, countries involved in VPA exported more to the US than to the EU.

## 6. Results

### 6.1. Analysis Results of Policy Effects for Coniferous Lumber

Table 3 presents the analysis outcomes regarding the policy effects of the EUTR and the Lacey Act. The findings reveal that the introduction of the timber legality requirement system reduced coniferous lumber imports from both the EU and the US. Specifically, the policy effect of each system on coniferous lumber imports ranged from −0.32% to −0.05% for EUTR and from −0.93% to −0.09% for the Lacey Act.

In addition, the Lacey Act exhibited a relatively more pronounced influence in reducing import volumes compared to the EUTR concerning coniferous lumber imports.

Table 4 simultaneously analyzes the policy effects of the VPA along with those of the EUTR and the Lacey Act. The estimation results for the gravity variable indicate that as trade distance increased by 1%, coniferous timber imports decreased by between −0.64% and −0.14% in the EUTR model, and by between −0.37% and −0.07% in the Lacey Act model, with the sign of the trade distance variable aligning with expectations.

Regarding GDP per capita and population, the EUTR model reveals that a 1% increase in the importing country's GDP per capita corresponds to a coniferous lumber import volume increase ranging from 0.06% to 0.64%. Similarly, a 1% rise in the importing country's population led to a coniferous lumber import volume increase ranging from 0.08% to 0.64%.

The results of the analysis of purchasing power and economic size of coniferous lumber-exporting countries yielded mixed results, with variations depending on the analysis method. While some findings indicated a decrease in exports as purchasing power and economic size increased, others pointed to an increase. A previous study argues that as the

purchasing power and economic size of an exporting country increase, the consumption of the exporting country increases, and consequently, the export volume decreases [37]. Conversely, studies have suggested that the increase in export volume can be attributed to the growing purchasing power and market size of the exporting country. This growth leads to an increased purchase volume of logs, which are the raw materials for exported lumber, consequently boosting the overall volume of lumber exports [40,41].

The study also considered the construction economy of importing countries, using the proportion of production in the construction industry as a proxy variable. The analysis revealed that a 1% increase in the proportion of production in the construction industry in the EU and the US led to coniferous lumber import volume increases ranging from 0.11% to 0.96% for the EUTR model and 0.11% to 0.52% for the Lacey Act model.

**Table 3.** Results of the policy effect analysis of the EUTR and the Lacey Act on coniferous lumber.

| | EUTR (Due-Diligence) | | | Lacey Act (Due-Care) | | |
|---|---|---|---|---|---|---|
| | Robust OLS (RSE) | Fixed Effect ($t$-Value) | PPML ($z$-Value) | Robust OLS (RSE) | Fixed Effect ($t$-Value) | PPML ($z$-Value) |
| $\ln\left(Dist_{i,j}\right)$ | −0.65 *** (0.01) | (omitted) | −0.14 *** (−68.93) | −0.39 *** (0.03) | (omitted) | −0.07 *** (−16.41) |
| $\ln(GDP\ capita_i)$ | 0.32 *** (0.01) | 0.64 *** (19.43) | 0.06 *** (27.66) | 0.30 *** (0.02) | 0.71 *** (13.21) | 0.06 *** (19.12) |
| $\ln\left(GDP\ capita_j\right)$ | 0.23 *** (0.01) | −0.55 *** (−17.62) | −0.01 *** (−9.32) | 0.31 *** (0.02) | −0.48 *** (−9.64) | 0.06 *** (16.05) |
| $\ln(POP_i)$ | 0.36 *** (0.01) | 1.07 *** (9.08) | 0.08 *** (59.07) | 0.28 *** (0.01) | 1.79 *** (9.82) | 0.04 *** (15.68) |
| $\ln\left(POP_j\right)$ | 0.07 *** (0.01) | −2.27 *** (−13.83) | 0.04 *** (28.00) | 0.20 *** (0.01) | −1.62 *** (−6.26) | 0.05 *** (21.32) |
| $\ln(GDPCons_i)$ | 0.58 *** (0.05) | 0.86 *** (9.20) | 0.14 *** (18.06) | 0.51 *** (0.07) | 0.19 (1.37) | 0.10 *** (8.00) |
| $(t_{2013})$ | −0.05 (0.04) | 0.31 *** (11.43) | −0.02 *** (−3.36) | - | - | - |
| $(t_{2008})$ | - | - | - | −0.48 *** (0.05) | −0.13 ** (−2.56) | −0.09 *** (−10.25) |
| $treat_{EU,j}$ | 0.42 *** (0.05) | (omitted) | 0.13 *** (19.33) | - | - | - |
| $treat_{Lacey,j}$ | - | - | - | 0.53 *** (0.19) | (omitted) | 0.07 ** (2.58) |
| $\left(treat_{EU,j} \times t_{2013}\right)$ | −0.16 ** (0.08) | −0.32 *** (−7.05) | −0.05 *** (−4.86) | - | - | - |
| $\left(treat_{Lacey,j} \times t_{2008}\right)$ | - | - | - | −0.64 ** (0.28) | −0.93 *** (−6.81) | −0.09 ** (−2.32) |
| $Cons$ | 0.87 *** (0.25) | 11.40 *** (8.56) | 0.88 *** (24.19) | −1.63 *** (0.32) | 3.09 (1.46) | 0.20 *** (3.22) |
| Obs. | | 40,666 | | | 17,071 | |
| F-test (Pseudo LL) | 611.41 *** | 141.63 *** | (−139,837) | 136.5 *** | 55.0 *** | (−42,808) |
| $R^2$ (within $R^2$) | 0.11 | (0.03) | - | 0.07 | (0.03) | - |
| Hausman test | - | 703.46 *** | - | - | 221.26 *** | - |

Note—***: significance level < 1%; **: significance level < 5%. RSE: Robust Standard Error; LL: Log-likelihood.

**Table 4.** Results of simultaneous effects of EUTR and Lacey Act with VPA for coniferous lumber.

| | EUTR (Due-Diligence) | | | Lacey Act (Due-Care) | | |
|---|---|---|---|---|---|---|
| | Robust OLS (RSE) | Fixed Effect (*t*-Value) | PPML (*z*-Value) | Robust OLS (RSE) | Fixed Effect (*t*-Value) | PPML (*z*-Value) |
| $\ln\left(Dist_{i,j}\right)$ | −0.64 *** (0.01) | (omitted) | −0.14 *** (−71.85) | −0.37 *** (0.03) | (omitted) | −0.07 *** (−15.46) |
| $\ln(GDP\ capita_i)$ | 0.34 *** (0.01) | 0.63 *** (18.59) | 0.00 (0.74) | 0.31 *** (0.02) | 0.66 *** (12.17) | 0.06 *** (16.20) |
| $\ln\left(GDP\ capita_j\right)$ | 0.21 *** (0.01) | −0.53 *** (−16.75) | 0.06 *** (28.75) | 0.28 *** (0.02) | −0.47 *** (−9.31) | 0.05 *** (16.38) |
| $\ln(POP_i)$ | 0.37 *** (0.01) | 1.01 *** (8.46) | 0.04 *** (27.45) | 0.28 *** (0.01) | 1.55 *** (8.22) | 0.05 *** (21.21) |
| $\ln\left(POP_j\right)$ | 0.07 *** (0.01) | −2.27 *** (−13.57) | 0.09 *** (61.84) | 0.21 *** (0.01) | −1.75 *** (−6.58) | 0.04 *** (15.70) |
| $\ln(GDPCons_i)$ | 0.56 *** (0.05) | 0.92 *** (9.59) | 0.13 *** (17.47) | 0.52 *** (0.07) | 0.40 *** (2.73) | 0.11 *** (8.23) |
| $(t_{2008})$ | - | - | - | −0.52 *** (0.07) | −0.19 *** (−3.42) | −0.10 *** (−6.81) |
| $(t_{2010})$ | −0.24 *** (0.06) | 0.05 (1.37) | −0.06 *** (−6.46) | 0.10 (0.11) | 0.13 ** (1.97) | 0.02 (0.82) |
| $(t_{2011})$ | −0.08 (0.08) | −0.06 (−1.48) | 0.00 (−0.16) | −0.15 (0.11) | −0.08 (−1.09) | −0.03 (−1.25) |
| $(t_{2013})$ | 0.07 (0.08) | 0.19 *** (4.17) | 0.01 (0.52) | 0.10 (0.12) | 0.17 ** (2.45) | 0.02 (0.82) |
| $(t_{2014})$ | 0.16 ** (0.07) | 0.19 *** (4.82) | 0.02 ** (2.11) | 0.09 (0.11) | 0.04 (0.60) | 0.02 (1.03) |
| $treat_{(EU,j)}$ | 0.38 *** (0.05) | (omitted) | 0.12 *** (18.29) | - | - | - |
| $treat_{(Lacey,j)}$ | - | - | - | 0.51 *** (0.19) | (omitted) | 0.07 ** (2.49) |
| $treat_{(i,GHA)}$ | −1.10 *** (0.18) | (omitted) | 0.28 *** (14.61) | −0.55 ** (0.24) | (omitted) | −0.12 * (−1.74) |
| $treat_{(i,CMR)}$ | −1.01 *** (0.16) | (omitted) | 0.33 *** (18.09) | −0.70 *** (0.22) | (omitted) | −0.17 ** (−2.48) |
| $treat_{(i,COG)}$ | −0.84 *** (0.26) | (omitted) | 0.17 *** (7.86) | −0.70 ** (0.31) | (omitted) | −0.21 ** (−2.02) |
| $treat_{(i,IDN)}$ | −0.57 *** (0.12) | (omitted) | 0.24 *** (15.15) | −0.71 *** (0.14) | (omitted) | −0.14 *** (−3.87) |
| $\left(treat_{(EU,j)} \times t_{2013}\right)$ | −0.16 * (0.08) | −0.32 *** (−7.16) | −0.05 *** (−4.88) | - | - | - |
| $\left(treat_{(Lacey,j)} \times t_{2008}\right)$ | - | - | - | −0.60 ** (0.27) | −0.93 *** (−6.82) | −0.09 ** (−2.14) |
| $\left(treat_{(i,GHA)} \times t_{2010}\right)$ | 0.13 (0.24) | 0.38 (1.60) | −0.14 *** (−4.62) | −0.27 (0.30) | −0.17 (−0.60) | −0.07 (−0.71) |
| $\left(treat_{(i,CMR)} \times t_{2011}\right)$ | −0.33 (0.25) | −0.09 (−0.31) | 0.01 (0.50) | −0.54* (0.30) | −0.09 (−0.26) | −0.14 (−1.23) |
| $\left(treat_{(i,COG)} \times t_{2013}\right)$ | −0.83 ** (0.41) | −0.33 (−0.64) | 0.10 *** (2.77) | −0.68 (0.42) | −0.43 (−0.80) | −0.16 (−0.80) |
| $\left(treat_{(i,IDN)} \times t_{2014}\right)$ | −1.16 *** (0.27) | −1.03 *** (−4.16) | −0.21 *** (−5.71) | −1.22 *** (0.30) | −1.26 *** (−4.43) | −0.27 *** (−2.78) |

**Table 4.** *Cont.*

| | EUTR (Due-Diligence) | | | Lacey Act (Due-Care) | | |
|---|---|---|---|---|---|---|
| | Robust OLS (RSE) | Fixed Effect (*t*-Value) | PPML (*z*-Value) | Robust OLS (RSE) | Fixed Effect (*t*-Value) | PPML (*z*-Value) |
| *Cons* | 0.83 *** (0.25) | 11.65 *** (8.45) | 0.76 *** (20.69) | −1.57 *** (0.32) | 5.64 ** (2.51) | 0.21 *** (3.48) |
| Obs. | | 40,666 | | | 17,071 | |
| F-test (Pseudo LL) | 298.39 *** | 74.09 *** | (−139,203) | 69.49 *** | 28.82 *** | (−42,744.7) |
| $R^2$ (within $R^2$) | 0.11 | (0.03) | - | 0.08 | (0.03) | - |
| Hausman test | - | 637.44 *** | - | - | 226.44 *** | - |

Note—***: significance level < 1%; **: significance level < 5%; *: significance level < 10%. RSE: Robust Standard Error; LL: Log-likelihood.

The policy effect of the timber legality requirement system indicates that the EUTR reduced coniferous lumber imports by between −0.32% and −0.05%, while the Lacey Act resulted in reductions ranging from −0.93% to −0.09%. These results were not significantly different from the previously analyzed estimated values presented in Table 3.

Specifically, for Ghana (GHA) and Indonesia (IND), both VPA-involved countries, coniferous lumber exports to the EU decreased, with Ghana experiencing a 0.14% reduction and Indonesia showing a decline ranging from 0.21% to 1.16%. By contrast, the Republic of Congo (COG) exhibited both an increase and decrease in coniferous lumber exports, depending on the analysis method. However, the effect of Cameroon's VPA was not statistically significant. In addition, coniferous lumber exports to the US from Cameroon (CMR) and Indonesia (IND) decreased by −0.54%, and by between −1.26% and −0.27%, respectively. Statistically significant results were not obtained for either Ghana or the Republic of Congo.

### 6.2. Analysis Results of Policy Effects for Non-Coniferous Wood

Table 5 analyzes the policy effects of the EUTR and the Lacey Act on non-coniferous lumber. The analysis demonstrates that the implementation of the EUTR resulted in decreased imports of non-coniferous lumber, ranging from −0.45% to −0.05%. By contrast, the policy effect of the Lacey Act was not statistically significant.

These findings align with the earlier observed consistent decrease in the import volume of both coniferous and non-coniferous lumber in the EUTR model. However, while the Lacey Act previously reduced the import volume of coniferous lumber, it did not exhibit statistical significance in influencing non-coniferous lumber imports. In other words, the Lacey Act did not have a discernible policy effect on non-coniferous lumber imports.

Table 6 simultaneously analyzes the policy effects of the EUTR, Lacey Act, and VPA. The gravity variable analysis yielded the following results: when trade distance increased by 1% for the dependent variable, the import volume of non-coniferous lumber, the EUTR model indicated a decrease in import volume ranging from −0.67% to −0.14%, while the Lacey Act model reduced imports by between −0.54% and −0.12%. These results mirror those obtained for coniferous lumber and align with the expected coefficient signs. The analysis of GDP per capita and population showed that a 1% increase in the importing country's GDP per capita led to import volume increases ranging from 0.06% to 0.51% in the EUTR model and from 0.06% to 0.66% in the Lacey Act model. Similarly, a 1% rise in the importing country's population corresponded to import volume increases ranging from 0.09% to 0.41% in the EUTR model and from 0.09% to 0.39% in the Lacey Act model. The GDP per capita of timber-exporting countries was positively related to the import volume of non-coniferous lumber, excluding the analysis results using the Fixed Effect model. The signs presented in the analysis results of the non-coniferous lumber import volume model were consistent with previous results for coniferous lumber.

**Table 5.** Results of the policy effect analysis of EUTR and Lacey Act on non-coniferous lumber.

| | EUTR (Due-Diligence) | | | Lacey Act (Due-Care) | | |
|---|---|---|---|---|---|---|
| | Robust OLS (RSE) | Fixed Effect (*t*-Value) | PPML (*z*-Value) | Robust OLS (RSE) | Fixed Effect (*t*-Value) | PPML (*z*-Value) |
| $\ln\left(Dist_{i,j}\right)$ | −0.64 *** (0.01) | (omitted) | −0.14 *** (−68.93) | −0.50 *** (0.02) | (omitted) | −0.11 *** (−32.68) |
| $\ln(GDP\ capita_i)$ | 0.24 *** (0.01) | 0.48 *** (20.00) | 0.06 *** (27.66) | 0.27 *** (0.01) | 0.66 *** (17.96) | 0.02 *** (11.70) |
| $\ln\left(GDP\ capita_j\right)$ | −0.04 *** (0.01) | −0.39 *** (−18.05) | 0.00 *** (−9.32) | 0.11 *** (0.01) | −0.26 *** (−7.61) | 0.06 *** (21.67) |
| $\ln(POP_i)$ | 0.36 *** (0.01) | −0.08 (−0.89) | 0.08 *** (59.07) | 0.37 *** (0.01) | 0.07 (0.56) | 0.04 *** (26.22) |
| $\ln\left(POP_j\right)$ | 0.18 *** (0.01) | −1.16 *** (−12.48) | 0.04 *** (28.00) | 0.20 *** (0.01) | −0.47 *** (−3.28) | 0.08 *** (42.85) |
| $\ln(GDPCons_i)$ | 0.55 *** (0.03) | 0.85 *** (11.40) | 0.14 *** (18.06) | 0.49 *** (0.05) | 0.25 ** (2.41) | 0.12 *** (11.69) |
| $(t_{2013})$ | −0.10 *** (0.02) | 0.17 *** (8.40) | −0.02 *** (−3.36) | - | - | - |
| $(t_{2008})$ | - | - | - | −0.47 *** (0.03) | −0.31 *** (−8.36) | −0.11 *** (−16.17) |
| $treat_{EU,j}$ | 0.62 *** (0.03) | (omitted) | 0.13 *** (19.33) | - | - | - |
| $treat_{Lacey,j}$ | - | - | - | 0.31 ** (0.12) | (omitted) | 0.04 * (1.80) |
| $\left(treat_{(EU,j)} \times t_{2013}\right)$ | −0.28 *** (0.05) | −0.45 *** (−13.71) | −0.05 *** (−4.86) | - | - | - |
| $\left(treat_{(Lacey,j)} \times t_{2008}\right)$ | - | - | - | −0.05 (0.16) | 0.07 (0.78) | 0.01 (0.28) |
| *Cons* | 2.73 *** (0.17) | 10.87 *** (12.63) | 0.88 *** (24.19) | 0.12 (0.22) | 3.96 *** (3.11) | 0.37 *** (7.89) |
| Obs. | | 60,132 | | | 30,126 | |
| F-test (Pseudo LL) | 1136.47 *** | 198.99 *** | (−139,837) | 378.74 *** | 57.48 *** | (−68,895) |
| $R^2$ (within $R^2$) | 0.14 | (0.03) | - | 0.11 | (0.02) | - |
| Hausman test | - | 553.65 *** | - | - | 171.11 *** | - |

Note—***: significance level < 1%; **: significance level < 5%; *: significance level < 10%. RSE: Robust Standard Error; LL: Log-likelihood.

The proportion of production in the construction industry in the importing country also had an impact. When this proportion increased by 1%, non-coniferous lumber imports rose by between 0.14% and 0.85% in the EUTR model and by between 0.12% and 0.49% in the Lacey Act model, with the EUTR model indicating relatively larger imports related to the construction industry. These results are similar to those previously reported for coniferous lumber.

In terms of policy effects, the EUTR was found to reduce the import volume of non-coniferous lumber by between –0.05% and –0.44%, while the Lacey Act did not achieve statistical significance. The policy effects of the EUTR and the Lacey Act in Table 6 show similar estimates and results to those in Table 5.

**Table 6.** Results of simultaneous effects of EUTR, Lacey Act with VPA for non-coniferous lumber.

| | EUTR (Due-Diligence) | | | Lacey Act (Due-Care) | | |
|---|---|---|---|---|---|---|
| | **Robust OLS (RSE)** | **Fixed Effect (*t*-Value)** | **PPML (*z*-Value)** | **Robust OLS (RSE)** | **Fixed Effect (*t*-Value)** | **PPML (*z*-Value)** |
| $\ln\left(Dist_{i,j}\right)$ | −0.67 *** (0.01) | (omitted) | −0.14 *** (−71.85) | −0.54 *** (0.02) | (omitted) | −0.12 *** (−35.21) |
| $\ln(GDP\,capita_i)$ | 0.26 *** (0.01) | 0.51 *** (20.44) | 0.06 *** (28.75) | 0.27 *** (0.01) | 0.66 *** (17.70) | 0.06 *** (22.15) |
| $\ln\left(GDP\,capita_j\right)$ | 0.03 *** (0.01) | −0.34 *** (−14.91) | 0.00 (0.74) | 0.18 *** (0.01) | −0.22 *** (−6.03) | 0.04 *** (17.33) |
| $\ln(POP_i)$ | 0.38 *** (0.01) | −0.02 (−0.26) | 0.09 *** (61.84) | 0.39 *** (0.01) | 0.10 (0.81) | 0.09 *** (44.67) |
| $\ln\left(POP_j\right)$ | 0.18 *** (0.01) | −1.33 *** (−13.65) | 0.04 *** (27.45) | 0.20 *** (0.01) | −0.77 *** (−4.91) | 0.05 *** (25.51) |
| $\ln(GDPCons_i)$ | 0.52 *** (0.03) | 0.81 *** (10.68) | 0.13 *** (17.47) | 0.49 *** (0.05) | 0.27 ** (2.43) | 0.12 *** (11.69) |
| $(t_{2008})$ | - | - | - | −0.47 *** (0.05) | −0.30 *** (−7.77) | −0.11 *** (−10.41) |
| $(t_{2010})$ | −0.33 *** (0.04) | −0.13 *** (−4.58) | −0.06 *** (−6.46) | −0.06 (0.07) | −0.04 (−0.92) | −0.01 (−0.95) |
| $(t_{2011})$ | 0.00 (0.05) | 0.01 (0.28) | 0.00 (−0.16) | 0.01 (0.07) | 0.03 (0.62) | 0.01 (0.34) |
| $(t_{2013})$ | 0.03 (0.05) | 0.08 *** (2.43) | 0.01 (0.52) | −0.08 (0.07) | −0.05 (−1.01) | −0.02 (−1.07) |
| $(t_{2014})$ | 0.11 ** (0.05) | 0.17 *** (5.65) | 0.02 ** (2.11) | 0.07 (0.06) | 0.06 (1.31) | 0.02 (1.17) |
| $treat_{(EU,j)}$ | 0.59 *** (0.03) | (omitted) | 0.12 *** (18.29) | - | - | - |
| $treat_{(Lacey,j)}$ | - | - | - | 0.33 *** (0.12) | (omitted) | 0.04 ** (1.97) |
| $treat_{(i,GHA)}$ | 1.35 *** (0.08) | (omitted) | 0.28 *** (14.61) | 1.25 *** (0.10) | (omitted) | 0.26 *** (10.78) |
| $treat_{(i,CMR)}$ | 1.63 *** (0.09) | (omitted) | 0.33 *** (18.09) | 1.05 *** (0.10) | (omitted) | 0.23 *** (9.74) |
| $treat_{(i,COG)}$ | 0.75 *** (0.08) | (omitted) | 0.17 *** (7.86) | 0.36 *** (0.10) | (omitted) | 0.08 ** (2.59) |
| $treat_{(i,IDN)}$ | 1.18 *** (0.08) | (omitted) | 0.24 *** (15.15) | 0.99 *** (0.10) | (omitted) | 0.20 *** (10.16) |
| $\left(treat_{EU,j} \times t_{2013}\right)$ | −0.28 *** (0.05) | −0.44 *** (−13.36) | −0.05 *** (−4.88) | - | - | - |
| $\left(treat_{Lacey,j} \times t_{2008}\right)$ | - | - | - | −0.05 (0.15) | 0.08 (0.85) | 0.01 (0.25) |
| $\left(treat_{(i,GHA)} \times t_{2010}\right)$ | −0.69 *** (0.12) | −0.18 * (−1.88) | −0.14 *** (−4.62) | −0.56 *** (0.15) | −0.18 (−1.51) | −0.11 *** (−3.01) |
| $\left(treat_{(i,CMR)} \times t_{2011}\right)$ | 0.01 (0.14) | 0.66 *** (7.15) | 0.01 (0.50) | 0.42 *** (0.15) | 0.75 *** (6.71) | 0.08 ** (2.46) |
| $\left(treat_{(i,COG)} \times t_{2013}\right)$ | 0.46 *** (0.14) | 1.04 *** (9.14) | 0.10 ** (2.77) | 0.68 *** (0.16) | 0.91 *** (6.54) | 0.15 *** (3.46) |
| $\left(treat_{(i,IDN)} \times t_{2014}\right)$ | −1.03 *** (0.18) | −0.72*** (−6.40) | −0.21 *** (−5.71) | −0.89 *** (0.22) | −0.71 *** (−5.31) | −0.18 *** (−4.06) |

**Table 6.** *Cont.*

| | EUTR (Due-Diligence) | | | Lacey Act (Due-Care) | | |
|---|---|---|---|---|---|---|
| | **Robust OLS (RSE)** | **Fixed Effect (*t*-Value)** | **PPML (*z*-Value)** | **Robust OLS (RSE)** | **Fixed Effect (*t*-Value)** | **PPML (*z*-Value)** |
| *Cons* | 2.17 *** (0.17) | 11.14 *** (12.39) | 0.76 *** (20.69) | −0.28 (0.22) | 5.56 *** (4.02) | 0.27 *** (5.85) |
| Obs. | | 60,132 | | | 30,126 | |
| F-test (Pseudo LL) | 576.30 *** | 117.04 *** | (−139,203) | 186.97 *** | 34.98 *** | (−68,616) |
| $R^2$ (within$R^2$) | 0.15 | (0.03) | - | 0.13 | (0.02) | - |
| Hausman test | - | 931.22 *** | - | - | 203.06 *** | - |

Note—***: significance level < 1%; **: significance level < 5%; *: significance level < 10%. RSE: Robust Standard Error; LL: Log-likelihood.

The policy effects for the VPA countries are as follows. The countries whose exports of non-coniferous lumber to the EU increased under the VPA were Cameroon (CMR) and the Republic of Congo (COG). Compared to countries that did not sign the VPA, Cameroon (CMR) witnessed a 0.66% rise in its non-coniferous lumber exports, while the Republic of Congo (COG) experienced an increase ranging from 0.10% to 1.04%. By contrast, Ghana (GHA) and Indonesia (IDN) showed a decrease in lumber exports compared to countries that did not sign the VPA.

Examining the export volume of non-coniferous lumber to the US among VPA countries revealed similar trends to those observed for the EU. Cameroon (CMR) experienced an export increase ranging from 0.08% to 0.75%, while the Republic of Congo (COG) witnessed an increase ranging from 0.15% to 0.91%. By contrast, Ghana's export volume decreased by between −0.69% and −0.14%, and that of Indonesia (IND) decreased by between −0.89% and −0.18%. Notably, Indonesia stood out as a VPA country where exports of both coniferous and non-coniferous lumber decreased.

## 7. Discussions

The main points of discussion in this paper are the following. Firstly, the policy effect of the Lacey Act on non-coniferous lumber was not observed. Secondly, since the period in which the Lacey Act was introduced coincides with the time when the US Financial Crisis took place, whether the policy effect of the Lacey Act was identified depends on whether the impact of the US Financial Crisis was appropriately controlled for.

First, the reason why the policy effect of the Lacey Act on non-coniferous lumber was statistically insignificant is as follows. The non-coniferous import quantities of the US are at a negligible level compared to those of the rest of the world. Therefore, it may have been more efficient to replace small quantities of imports of legally produced non-coniferous lumber with domestic non-coniferous lumber. In addition, it is difficult to rule out the impact of a trade diversion effect in which existing non-coniferous lumber exporting countries do not export to the US, which requires timber legality [20].

The introduction of the Lacey Act in 2008 coincides with the onset of the US Financial Crisis. Therefore, the difference-in-differences method is an appropriate method in terms of controlling for the impact of the US Financial Crisis and identifying the net effect of the Lacey Act. The US Financial Crisis is deemed to have reduced worldwide aggregate demand [42]. If the policy effect is analyzed based on its implementation in 2008 without comparing the US (treatment group) to countries other than the US (control group), there is a strong likelihood that the impact of the financial crisis is included. However, since the difference-in-differences method compares the treatment and control groups based on the implementation of the Lacey Act in 2008, its policy effect can be seen as independent from the impact of the US Financial Crisis.

Meanwhile, the analysis results of the policy effect of the VPA, which points to the decrease in timber exports from countries that signed the VPA, are as follows. Firstly, because the VPA is based on Due-diligence, the timber legality requirement system requires the legality of logs in the supply chain, which are raw materials. Therefore, if a VPA country cannot import legal logs, it is also impossible for it to export lumber that was produced using the imported raw materials to the EU and the US, which have also introduced a timber legality requirement system. Secondly, the implementation of Due-diligence as part of the VPA incurs administrative and management costs. From an economic perspective, one of the problems with illegal timber is price competitiveness due to failure to appropriately pay taxes in accordance with the laws of the producing country. However, since the implementation of Due-diligence requires various costs, price competitiveness is relatively less intense compared to the state before the introduction of the VPA. Indonesia has determined that the competitiveness of its timber industry has decreased after the implementation of the VPA in the EU timber market [43]. Moreover, even if timber is produced and exported legally through the VPA, the market does not provide compensation in the form of a price premium for legally produced wood [44,45]. In particular, from the perspective of timber-exporting countries that import and process coniferous wood, the VPA will act as a significant factor in reducing the quantity of coniferous lumber exports.

The limitations of this study are, first, that the temporal range of data used in the analysis did not include recent data, due to the temporary stoppage of the FAO's provision of panel data from 2018 to the present. If the panel data are updated in the future, it will be possible to use it to analyze recent international issues using the model. Second, in order to ensure the robustness of the policy effect through the difference-in-differences method, it is necessary to test the parallel trend hypothesis. To achieve this, a country with similar timber trade quantities and economic size must be selected as a control group. However, there is a limitation in that realistic control groups for the EU and the US cannot be found. Third, the policy effect of the Lacey Act for non-coniferous lumber did not show statistical significance. Therefore, a clear identification of the cause of this result is necessary. This requires an approach using qualitative methods, such as interviews with competent authorities.

## 8. Conclusions

The timber legality requirement system provides a powerful policy tool to prohibit the importation of illegal timber. Therefore, in order to justify and rationalize the implementation of this system, it is necessary to quantitatively investigate the policy effect of this system, which prohibits trade in illegal timber. To this end, this study analyzed the policy effects of the timber legality requirement system and the VPA in the trade of coniferous and non-coniferous lumber.

The analysis results of the policy effect of the timber legality requirement system by applying the difference-in-differences method using the gravity model suggest that the US and EU saw a decrease in lumber imports after the introduction of the timber legality requirement system. Additionally, even with the implementation of the VPA, which encouraged legal production of lumber, lumber exports were found to have decreased. Existing studies report that among timber-exporting countries, the proportion of timber produced illegally in certain countries reaches 10%–15% [46]. Therefore, since the timber legality requirement system only allows the import of legally produced timber, it suggests that the result of reduced lumber imports following the introduction of the timber legality requirement system is valid.

Coniferous lumber exports decreased in most countries that signed the VPA. These results are in contrast to previous studies that found no clear difference in export volume of non-coniferous lumber between the VPA and non-VPA countries. In addition, the trade diversion effect of VPA countries exporting lumber to countries that do not require timber legality has been mentioned in the literature [13]. If the trade diversion effect of VPA countries is observed, it suggests that there is no difference in lumber export volume

between VPA and non-VPA countries. Therefore, statistical significance of the policy effect would not be obtained. However, the fact that lumber exports from VPA countries have decreased compared to non-VPA countries suggests that producing illegal timber under the Due-diligence on a national level may be rather inefficient. Therefore, the significance of the results of this study lies in the fact that it shows the possibility of the VPA's ability to control the trade diversion effect of timber-producing countries.

Meanwhile, the reason why timber-exporting countries reach a VPA with the EU is to facilitate trade or expand exports without restricting imports from countries with large markets that require timber legality, such as the EU and the US. However, the decline in timber exports from VPA countries has significant implications for timber-exporting countries involved in VPA negotiations with the EU. Additionally, if the lumber trade volume decreases due to the introduction of a timber legality requirement system, import prices may rise, which may have a negative impact on the social welfare of timber-importing countries. Therefore, in order to stably produce and export legal timber through the VPA, it is necessary for the EU to make efforts to ensure smooth trade of timber by providing support for timber-exporting countries to efficiently implement the VPA.

Future research is needed to analyze the policy effect of the timber legality requirement system for wood products such as plywood and fiberboard, which have complex supply chains and are produced by importing raw materials from various countries. Furthermore, in addition to the EU and the US, major timber-demanding countries such as Australia, Japan, Korea, and China are also introducing timber legality requirement systems. Therefore, it is necessary to conduct further analysis targeting these countries.

**Author Contributions:** Conceptualization, D.-H.K.; methodology, D.-H.K.; software, D.-H.K.; validation, K.-D.K., H.-I.C. and G.S.; formal analysis, D.-H.K.; writing—original draft preparation, D.-H.K., K.-D.K. and H.-I.C.; writing—review and editing, G.S.; visualization, D.-H.K. All authors have read and agreed to the published version of the manuscript.

**Funding:** This study received no external funding.

**Data Availability Statement:** Data available in a publicly accessible repository.

**Conflicts of Interest:** The authors declare no conflict of interest.

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
