# Peer review of "Effect of the Timber Legality Requirement System on Lumber Trade: Focusing on EUTR and Lacey Act"

_forests, doi:10.3390/f14112232_

Round 1
Reviewer 1 Report (Previous Reviewer 1)
Comments and Suggestions for Authors
Dear authors,
I would like to inform you that I have reviewed the revised manuscript titled "Effect of the Timber Legality Requirement System on Timber Trade: Focusing on EUTR and Lacey Act," which you recently sent for my review.
After careful examination of the revised manuscript and the authors' responses to my previous comments, I would like to highlight that the authors have made significant improvements in line with the feedback. The revised manuscript now exhibits higher quality and clarity, making it ready for publication.
Therefore, I accept this revised version of the manuscript for publication. I appreciate the authors' efforts in enhancing their work and their cooperation throughout the review process.
Reviewer 2 Report (Previous Reviewer 2)
Comments and Suggestions for Authors
Dear authors,
thank you for considering my suggestions.
After your revisions, I accept you paper to be published.
This manuscript is a resubmission of an earlier submission. The following is a list of the peer review reports and author responses from that submission.
Round 1
Reviewer 1 Report
Comments and Suggestions for Authors
1. Introduction: Please move the sentence (On the international lumber …. ) from row 133 to new row.
2. Figure 1: The x and y-axis labels are missing on the graphs. The font in the graphs is too small and unreadable.
3. The study's use of data from 1997 to 2017 in its gravity model analysis is noteworthy, but it missed an opportunity to provide more up-to-date insights by not incorporating available data up to 2021, potentially limiting the relevance of its findings in the rapidly evolving timber trade landscape, especially in light of recent geopolitical events. In the description of the International Timber Trade Situation, data up to 2021 are provided but are not included in the model. We kindly request the authors to extend the model to include data up to 2021.
In light of the review, the manuscript shows promise but requires minor revisions, particularly concerning the incorporation of more recent data in the analysis. Addressing these concerns will significantly improve the quality and relevance of the research. Overall, the article has a strong foundation and the potential to make a valuable contribution to the field with these adjustments.
Reviewer 2 Report
Comments and Suggestions for Authors
I thank the authors for presenting a topic that is extremely interesting. However, I am not convinced that the content of the article and the source data were chosen appropriately so that significant conclusions could be drawn from the study.
1) combination of EUTR and the Lacey Act, which in principle addresses a broader area (wildlife, fish, plants).
2) What lumber do the authors mean? tropical or all lumber?
3) EUTR is one of the key terms in the paper, however authors do not refer to the content of EUTR (nearly at all).
4) missing "Discussion" as subchapter, e.g. including taking into account, among other things, the Regulation (EU) 2023/1115 of the European Parliament and of the Council of 31 May 2023.
4) misleading information for the reader, e.g. why and how were the countries/regions selected? (see, e.g., lines 42 and 43 - "countries", such as the US and Europe. These are regions. Do authors mean European states and/or the EU?
5) I also believe that the authors could have chosen more appropriate source literature (see e.g. 1-3) where they refer to deforetation, but this is not a direct problem in EU countries.
6) I don't understand e.g. Figure 1, which refers to ITTO data - again in relation to lumber, when e.g. in the European area, lumber from tropical species is not a major lumber used (see comment above).
The theoretical approach and information are hard to follow, and that is why my evaluation is to reject the paper in the present form.
